# Synthesis of Optically Tunable and Thermally Stable PMMA–PVA/CuO NPs Hybrid Nanocomposite Thin Films

**DOI:** 10.3390/polym13111715

**Published:** 2021-05-24

**Authors:** Ahmad M. Alsaad, Ahmad A. Ahmad, Issam A. Qattan, Abdul-Raouf El-Ali, Shatha A. Al Fawares, Qais M. Al-Bataineh

**Affiliations:** 1Department of Physical Sciences, Jordan University of Science & Technology, P.O. Box 3030, Irbid 22110, Jordan; sema_just@yahoo.com (A.A.A.); qalbataineh@ymail.com (Q.M.A.-B.); 2Department of Physics, Khalifa University of Science and Technology, Abu Dhabi 127788, United Arab Emirates; issam.qattan@ku.ac.ae; 3Department of Physics, Yarmouk University, Irbid 21163, Jordan; abedali@yu.edu.jo (A.-R.E.-A.); shada.Alfawares@yahoo.com (S.A.A.F.)

**Keywords:** organic–inorganic blends, poly methylmethacrylate (PMMA), polyvinylalcohol (PVA), copper oxide nanoparticles (CuO NPs), optical characterization, thermal stability, surface morphology

## Abstract

We report the synthesis and comprehensive characterization of polymethylmethacrylate (PMMA)/polyvinylalcohol (PVA) polymeric blend doped with different concentrations of Copper oxide (CuO) nanoparticles (NPs). The PMMA–PVA/CuO nanocomposite hybrid thin films containing wt.% = 0%, 2%, 4%, 8%, and 16% of CuO NPs are deposited on glass substrates via dip-coating technique. Key optical parameters are measured, analyzed, and interpreted. Tauc, Urbach, Spitzer–Fan, and Drude models are employed to calculate the optical bandgap energy (*E*_g_) and the optoelectronic parameters of PMMA–PVA/CuO nanocomposites. The refractive index and *E*_g_ of undoped PMMA–PVA are found to be (1.5–1.85) and 4.101 eV, respectively. Incorporation of specific concentrations of CuO NPs into PMMA–PVA blend leads to a considerable decrease in *E*_g_ and to an increase of the refractive index. Moreover, Fourier Transform Infrared Spectroscopy (FTIR) transmittance spectra are measured and analyzed for undoped and doped polymeric thin films to pinpoint the major vibrational modes in the spectral range (500 and 4000 cm^−1^) as well as to elucidate the nature of chemical network bonding. Thermogravimetric analysis (TGA) is conducted under appropriate conditions to ensure the thermal stability of thin films. Doped polymeric thin films are found to be thermally stable below 105 °C. Therefore, controlled tuning of optoelectronic and thermal properties of doped polymeric thin films by introducing an appropriate concentration of inorganic fillers leads to a smart design of scaled multifunctional devices.

## 1. Introduction

The class of smart, functional materials based on organic–inorganic hybrid nanocomposites exhibit outstanding physical, chemical, thermal, and optical properties. Consequently, it plays a major role in the fabrication of modern scaled devices. Combining inorganic nanoparticles and polymers boosts the optical properties and alters the mechanical behavior of the resulting nanocomposite blend [1]. Utilizing a high refractive index material in optical applications frequently requires higher optical transparencies [1,2,3,4]. Polymers with high refractive index have fascinated several research groups owing to their potential applications in cutting-edge optoelectronic devices such as organic light-emitting diode devices [5], high-performance substrates for advanced display devices [6], antireflective coatings for advanced optical applications [7], and microlens components for charge-coupled devices or complementary metal oxide semiconductor [8]. Poly vinylalcohol (PVA) has been applied in the industrial, commercial, medical, and food sectors. It has been used to produce surgical threads, paper products, and food packaging materials. PVA has attracted considerable attention due to its attractive film-forming, good processability, biocompatibility, and good chemical resistance. It can be easily tarnished in water [9,10]. This polymer is widely used for blending with other polymer compounds, such as biopolymers and other polymers with hydrophilic properties. Owing to the enhanced structure, mechanical, and hydrophilic properties, the polymerized thin films can be utilized for various industrial applications. The addition of inorganic material to the polymeric matrix is advantageous to further enhance the chemical, structural, and physical properties [9]. PVA has OH groups arranged regularly from one side of the plane to the other, thus providing interchain hydrogen-bond networks. This may induce high optical clarity and a polarization response in the resulting hybrid polymerized thin films. Consequently, PVA polymer can be utilized in photovoltaic and optoelectronic devices [1]. Furthermore, Poly Methyl-Methacrylate (PMMA) can withstand temperatures between 70 up to 100 °C. Additionally, it possesses very good optical properties with a refractive index ranging between 1.3 and 1.7 [4]. Owing to its high impact strength, lightweight, and shatter resistance, the PMMA is one of the best organic optical materials, and it is widely used as a substitute for inorganic glass [11]. The PMMA polymer is selected for this study owing to several properties, such as its safety, chemical inertness, very good electrical properties, and excellent thermal stability. Additionally, it has been reported to be suitable for numerous imaging and non-imaging microelectronics, sensors, and conductive devices [12]. Copper oxide nanoparticles (CuO NPs) are black solid inorganic particles. CuO NPs exhibit a bandgap energy of 1.2 eV and adopts p-type conductivity due to the excess of oxygen or copper vacancies in their structure [13]. We have selected CuO NPs in this study based on their unique properties such as low cost, non-toxicity, ability to diffuse easily in polymers, good electronic and thermal properties, chemical stability, a high dielectric constant of 18.1, and a refractive index of 1.4 [13]. CuO NPs have been technologically implemented in photothermal and solar energy materials [13], supercapacitor [14], gas sensors, batteries [15], and optoelectronic devices. Nanoparticles could be as small as 1–100 nm. With such sizes, they exhibit limited mechanical, chemical and optical properties. Scaled industrialized multifunctional materials require entanglement of nanoparticles with organic components to yield distinguished nanocomposites. Such a blend exhibits multiphase segregation with phases that could be one, two or three dimensional or nanocomposite having nanoscale spaces periodically repeated between different phases. Such a simple approach leads to the fabrication of novel materials by utilizing different nanoscale building blocks to design and fabricate a new generation of optoelectronic devices with unique mechanical, chemical, thermal, vibrational, and optical properties. The novelty of this work can be clearly highlighted. The aspects of diffusing nanoparticles into the polymeric matrix are clearly apparent in PMMA–PVA/CuO NPs nanocomposite that other partners of the same family such as PMMA–PVA/TiO_2_ NPs and PMMA–PVA/SiO_2_ NPs. This could be attributed to the fact that phase segregation is clearly represented in this system. Using a simple approach to introduce CuO nanoparticles into PMMA–PVA polymer-matrix composites, distinguished mechanical, thermal, physical, electrical and optical properties are obtained. Integrating this work with previously published works on PMMA–PVA doped with other types of nanoparticles [16,17,18,19] is expected to provide the industrial market with a wide spectrum of novel materials having a wide range of distinguished optical, mechanical, and physical properties. Numerous efforts could then be performed to expand this approach for the fabrication of highly efficient devices.

The aim of this paper is two-fold. Firstly, PMMA–PVA/CuO NPs polymeric nanocomposite thin films are synthesized by dip coating on glass substrates. Secondly, key optical parameters of as-prepared doped and undoped thin films are measured and calculated by performing UV-Vis measurements and employing a combination of classical optical models. In particular, Transmittance (T %), Reflectance (R %), index of refraction (*n*), extinction coefficient (*k*), Urbach energy (*E_U_*), and optical bandgap energy (E_g_) are analyzed and interpreted. Moreover, we identify the major vibrational modes and appropriate conditions for the chemical stability of PMMA–PVA/CuO NPs nanocomposite thin films using FTIR and TGA techniques.

To elucidate the necessity of exploring the PMMA–PVA thin films doped with CuO NPs, we summarize the key optical and optoelectronic parameters of the PMMA–PVA/TiO_2_ and PMMA–PVA/SiO_2_ NPs nanocomposites published previously [17,19]. The significant influence of the type of the NPs introduced on the refractive index, high-frequency dielectric constant, extinction coefficient, optical resistivity, and optical mobility of the doped polymeric thin films motivates us to investigate the effect of introducing CuO NPs on the optical and optoelectronic properties of PMMA–PVA host polymer. Remarkably, introducing CuO NPs has demonstrated an incredible effect on the refractive index, optical resistivity, and optical mobility of PMMA–PVA thin films. This permits us to synthesize tunable and thermally stable polymeric thin films for the entire frequency spectrum. Careful examination of the values of optical and optoelectronic parameters of the three systems reveals that it is possible to synthesize a doped PMMA–PVA system that exhibits the essential properties for the fabrication of a new generation of low cost and efficient multifunctional scaled optoelectronic devices.

## 2. Materials and Methods

### 2.1. Materials

Copper (II) nitrate trihydrate Cu(NO_3_)_2_ with a molecular weight of 187.56 g/mol, hexamethylenetetramine [(CH_2_)_6_N_4_] (HMT) with a molecular wight of 140.186 g/mol, Poly(methylemethacrylate) (PMMA) [CH_2_=C[CH_3_]CO_2_H] with a molecular wight of 120,000 g/mol, Polyvinyl alcohol (PVA) with a molecular weight of 44.05 g/mol, and Chloroform with a molecular weight of 119.38 g/mol were purchased from Sigma-Aldrich (St. Louis, MO, USA).

### 2.2. Preparation of Copper Oxide Nanoparticles (CuO NPs)

The CuO NPs were prepared by employing the hydrothermal method. We mix 2 g of Copper (II) nitrate trihydrate Cu(NO_3_)_2_·3H_2_O with 0.5 g of hexamethylenetetramine [(CH_2_)_6_N_4_] (HMT) in a (beaker A). This white crystalline compound is highly soluble in water and polar organic solvents. It has a cage-like structure similar to adamantane. The contents of beaker A were dissolved in 50 mL of distilled water with a pH = 8. The solution is then placed in a hot bath at 90 °C for 5 h to obtain powdered CuO NPs. The obtained CuO NPs were washed three times with distilled water to remove impurities and residuals [9], centrifuged, and heated at temperatures ranging from 60 to 70 °C for 24 h. Finally, having obtained CuO NPs in the powder form, they were then placed in the furnace at a temperature of 400 °C for 2 h [20].

### 2.3. Preparation of Copper Oxide Nanoparticles (CuO NPs)

The glass substrates were cleaned with warm tap water, then rinsed with ionized acidic water (pH = 3.5) to remove the surface oxidized layer and greases and then dipped in acetone. The substrates were then bathed in an ultrasonic bath of distilled water for 5 min and dried with cold air. Then, 1 g of PMMA and 1 g of PVA were dissolved separately in 200 mL of chloroform [CHCl₃]. The PMMA–PVA solution was obtained by mixing PMMA and PVA solutions in a 1:1 volume ratio using a magnetic stirrer for 24 h to enhance the homogeneity. PMMA–PVA solution was then filtered using 0.45 μm Millipore filter before dip coating on the glass substrates. The films were synthesized at room temperature of 27 °C under atmospheric pressure [19]. The PVA polymers prevent the aggregation of the CuO NPs by the organic surface modification and keep the CuO NPs dispersed in the PVA matrix at the nanoscale. The final solutions were filtered by paper filter (0.45 μm in dimension). The viscosity of sol–gel solutions ranges between 1.2079 and 2.8935 Cp. The PMMA–PVA/CuO nanocomposite solutions were deposited as a thin layer on glass substrate using the dip-coating technique for 2 h, forming nanocomposite thin films with an average thickness of 500 nm with a maximum standard deviation of 7.5%. The thickness of thin films is confirmed by scanning electron spectroscopy (SEM) (Hitachi High-Tech America Inc., Life Sciences and Nanotechnology - Dallas, TX, USA). The nanocomposite thin films were obtained by allowing the solvent to evaporate for 15 min at 70 °C to evaporate the solvent and organic residues. The withdrawal speed ranged from 10 to 80 cm min^−1^. The multilayers of PMMA–PVA/CuO nanocomposites thin films were then analyzed and interpreted [21].

## 3. Results and Discussion

### 3.1. UV-Vis Spectroscopy

To elucidate the optical properties of PMMA–PVA/CuO NPs thin films, we analyzed the spectral behavior of transmittance (T%) and reflectance (R%) measured by A Double-Beam UV-Vis Spectrophotometer (U-3900H) (Hitachi High Technologies America, Inc., Pleasanton, California, USA) with an integrating sphere. Furthermore, related optical parameters such as absorption coefficient (α), the optical bandgap energy ($E_{g}$), Urbach energy ($E_{U}$), optical constants (n and k), and optical dielectric functions (ε₁ and ε₂) are measured, analyzed, and interpreted. By using Thermogravimetric Analysis (TGA) (NETZSCH Premier Technologies, Exton, PA, USA), we investigated the thermal stability. In addition, FTIR analysis is conducted to investigate the bonding and to identify the vibrational bands of the PMMA–PVA/CuO films. The FTIR results reveal strong interaction between the constituents of PMMA–PVA/CuO nanocomposite thin films as indicated by the induced changes in the vibration modes and the band positions.

Figure 1 shows the transmittance spectra of PMMA–PVA/CuO nanocomposite thin films. It clearly indicates an ion transfer mechanism between PMMA–PVA polymer composite and CuO NPs. The transmittance of undoped PMMA–PVA polymeric thin film in the visible region is found to be about 91.6%. Such value indicates an excellent excitation leads to a sharp electron transition from the valence band to the conduction band. Interestingly, introducing CuO NPs into PMMA–PVA polymeric thin films leads to a gradual and nonlinear decrease in transmittance in the visible region in agreement with the study of A. Abouhaswa and T. Taha [22] and A. Al-Hossainy [23]. Our results indicate that PMMA–PVA/CuO nanocomposite thin films with wt.% = 2% and 4% of CuO NPs exhibit transmittance of 77% and 72% at λ = 550 nm, respectively. Moreover, increasing the concentration of CuO NPs to wt.% = 8% and 16% leads to a decrease in the transmittance to 68% and 65% at λ = 550 nm, respectively. The loss of transparency is attributed to scattering by NPs within the polymeric matrix [24]. This occurs when the particle size is smaller than the wavelength, or it could happen as a result of electron transitions between polymer components and Cu^2+^ ions of the CuO NPs [25].

Figure 2 shows the reflectance spectra of PMMA–PVA/CuO nanocomposite thin films containing various CuO NPs. At λ≥ 350 nm, reflectance values slightly decrease as the wavelength is increased. The reflectance of undoped PMMA–PVA polymeric thin film is found to be about 4.9% at λ= 550 nm. Introducing wt.% = 2%, 4%, 8%, and 16% of CuO NPs into PMMA–PVA polymeric thin films leads to an increase of the reflectance to 6.5% in the visible region. The average sum of the transmittance and the reflectance of PMMA–PVA/CuO nanocomposite thin films is found to be less than one. This indicates the possibility of scattering or absorption of the incident light from constituents of PMMA–PVA/CuO nanocomposite thin films [26].

Figure 3 shows the absorbance coefficient (α) of PMMA–PVA/CuO nanocomposite thin films. Obviously, all PMMA–PVA/CuO nanocomposite thin films exhibit vanishingly small α in the visible region. However, α attains higher values in the UV region (λ ≤ 350 nm). Furthermore, introducing CuO NPs into PMMA–PVA polymeric matrix leads to an increase of α. The increase in α values may be attributed to the significant decrease in the interband transitions [24]. The PMMA–PVA/CuO nanocomposite thin films have strong absorption in the UV region and a strong transition in the visible region. Thin films of such features may act as key candidate potential materials for optoelectronic devices operating in the UV spectral region and can be employed as solar cell absorber or optical windows [24,27].

Figure 4 shows the refractive index (n) spectra of PMMA–PVA/CuO nanocomposite thin films for various CuO NPs concentrations. For the spectral region (λ<350 nm), incident light frequency exactly matched with the plasma frequency [24]. For λ≥350 nm, n decreases slightly as the wavelength is increased. Our results indicate that n of undoped PMMA–PVA polymeric thin film lies in the range (1.5–1.85) as the wavelength is decreased. Introducing wt.% of 2%, 4%, 8%, and 16% of CuO NPs into PMMA–PVA polymeric thin films leads to an increase of n to 1.68 at λ= 550 nm consistent with Q. Al Bataineh study [18]. This change is independent of the concentration level of CuO NPs incorporated in (PMMA–PVA) polymeric thin films. This increase in n value of PMMA–PVA/CuO nanocomposite thin films could be attributed to the condensation of smaller nanoparticles into larger clusters [26,28]. Consequently, reflectance is found to be independent of the doping level of the CuO NPs. The obtained fixed high values of n of PMMA–PVA/CuO nanocomposite thin films have a direct implication, as they can be employed in optoelectronic applications, such as optical waveguides, filters, lenses, antireflective coatings, and photonic device applications (solar cells and encapsulation materials for light-emitting diodes (LEDs)) [1,2,3,5,24,28,29].

The extinction coefficient (𝑘) measures the absorption loss. Specifically, it measures the fraction of light lost by scattering and absorption per unit distance of the medium [30]. It can be expressed as *k*=αλ/4π [30,31]. Figure 5 shows k spectra of PMMA–PVA/CuO nanocomposite thin films. Clearly, k  exhibits high values in the UV region and small values in the visible region. At λ= 550 nm, k increases to 0.0081, 0.0291, 0.0381, 0.0416, and 0.0417, respectively, as wt.% = 0%, 2%, 4%, 8%, and 16% of CuO NPs are added to PMMA–PVA polymeric thin films. The k parameter is directly proportional to α [25]. Therefore, such behavior of k indicates the presence of light absorption by PMMA–PVA/CuO nanocomposite thin films in the UV spectral region, as well as a considerable loss due to scattering in the visible light region [1]. Our results indicate that PMMA–PVA/CuO nanocomposite thin films exhibit significant loss of optical transparency in visible spectral regions [2].

The optical dielectric constant (ɛ′) indicates the suppression of the speed of light in the medium [32]. The complex dielectric function is $\varepsilon =\varepsilon^{\prime }+i\varepsilon^{\prime \prime },\;$ where ε’ is given by $\varepsilon ’=n^{2}+k^{2}$ [33]. The imaginary part is called optical dielectric loss $\left(\varepsilon ’’\right)$ and given by ε’’=2nk [34]. Figure 6 shows ε’ spectra of PMMA–PVA/CuO nanocomposite thin films as a function of wavelength for various CuO NPs concentrations. The value of ε’ of undoped PMMA–PVA thin film lies in the range (2.2–3.3). The insertion of wt.% = 2%, 4%, 8%, and 16% of CuO NPs into the PMMA–PVA matrix results in an increase of $\varepsilon^{\prime }$ to approximately 2.4 at λ = 550 nm. Remarkably, $\varepsilon^{\prime }\;$ of PMMA–PVA/CuO nanocomposite thin films demonstrates a uniform incremental increase, as CuO NPs contents in PMMA–PVA polymeric thin films are increased [1,26]. This trend may be attributed to the slight rotation of thermally activated dipoles [35]. The high dielectric constants are attributed to the relatively large polarizability of nonpolar bonds such as (C–H), (C–C), and (C=O) [36]. FTIR spectra measurements of thin films also support the presence of (C–H) and (C=O) bonds [1]. Figure 7 shows the optical dielectric loss (ε’’) spectra of PMMA–PVA/CuO nanocomposite thin films. Clearly, as-grown thin films exhibit high $\varepsilon^{\prime \prime }\;$ in the UV region. However, the PMMA–PVA/CuO nanocomposite thin films exhibit small values of ε’’ in the visible region. Therefore, less dissipation of energy from the electric field due to molecular dipoles motion in the visible region is observed [35]. Our results indicate that, at λ= 550 nm, optical dielectric loss exhibits values of 0.0076, 0.02915, 0.02916, 0.0416, and 0.0417, respectively, as wt.% = 0%, 2%, 4%, 8%, and 16% of CuO NPs are added into the PMMA–PVA matrix. Remarkably, (ε’ ≫ ε’’) indicating that PMMA–PVA/CuO NPs nanocomposites demonstrate extremely small dissipation of energy and higher speed of propagation of light. Table 1 summarizes key optical parameters of PMMA–PVA/CuO NPs nanocomposite thin films.

Tauc plot is based on relating the absorption coefficient with the incident photon energy (hv) as ${\left(\alpha hv\right)}^{2}=\beta \left(hv-E_{g}\right)$ [31,37,38,39], where β is constant called band tailing parameter, and $E_{g}$ is the band gap energy. It is obtained by plotting (hυ) in eV versus ${\left(\alpha h\upsilon \right)}^{2}$. Figure 8 shows the Tauc plot of the as-grown doped polymerized thin films. As shown, the optical bandgap energy $\left(E_{g}\right)$ values are obtained by extrapolating the linear part of the relation to the incident photon energy at which $E_{g}$ equals to the incident photon energy (hυ). The $E_{g}$ of undoped PMMA–PVA polymeric thin film is found to be 4.101 eV. As 2% and 4% of CuO NPs are introduced into the PMMA–PVA matrix, the value of $E_{g}$ decreases slightly to 4.0945 eV and 4.0853 eV, respectively. The injection of 8% and 16% of CuO NPs into PMMA–PVA/CuO nanocomposite thin films leads to a further slight decrease of $E_{g}$ to 4.0593 eV and 4.0374 eV, respectively, which is in good agreement with the study of A. Abouhaswa and T. Taha [22] and S. Ibrahim [40]. This may be attributed to the ability of CuO NPs to enhance the ion transfer between the constituents of PMMA–PVA/CuO nanocomposite thin films. Furthermore, some sub-bands are generated due to the formation of defect levels leading to facilitate the overpass of electrons from the valence band maximum (VBM) to the conduction band minimum (CBM) [26,41]. The increase in carrier density in localized states leads to a significant decrease in the transition probabilities of the extended states. As a result, additional absorption and reduction in the bandgap energy are observed [41].

Urbach’s energy $E_{U}$ is related to the local states broadened into the band gap predominantly due to the occurrence of defects, impurities, and non-crystallinity. The relation between $E_{U}$ and the absorption coefficient α is given by $\alpha =\alpha_{0}\exp\left(hv/E_{U}\right)$, where $\alpha_{0}$ is a constant [42,43]. $E_{U}$ is calculated by plotting lnα vs. hv. The reciprocal of the slopes of the linear portion, below the optical band gap, gives the value of $E_{U}$ [44]. Figure 9 shows the variation of Urbach energy $\left(E_{U}\right)$ of PMMA–PVA/CuO nanocomposites as a function of CuO NPs. $E_{U}$ of undoped PMMA–PVA film is found to be 190.7 meV. The insertion of wt.% = 2%, 4%, 8%, and 16% of CuO NPs increases $E_{U}$ to 211.1 meV, 224.8 meV, 242.5 meV, and 244.8 meV, respectively. In addition, the Urbach energy tail is closely related to the disorder in the film network. The increase in the values of Urbach energy as the concentration of CuO NPs contents in PMMA–PVA/CuO nanocomposites confirms the existence of defects and impurities [24].

Figure 10 shows the relation between $E_{g}$ and $E_{U}\;$ of PMMA–PVA/CuO films for various CuO NPs concentrations. Clearly, $E_{U}$ and the band edges of the PMMA–PVA/CuO films exhibit a reverse trend. It can be easily noticed that the lowering of the bandgap in PMMA–PVA/CuO nanocomposite thin films is due to the presence of localized defect states in the bandgap energy near the conduction band minimum and valence band maximum [24,41]. The decrease of $E_{U}$ reveals alterations in the structure since the Urbach tail parameter is a function of the structural disorder [1,45].

### 3.2. Optoelectronic Parameters

The frequency zone investigated determines the contribution of electronic, ionic, dipolar, and space charge polarization to the dielectric function. The influence of the space charge contribution is strongly dominant and basically responsible for normal dispersion in the low frequency region. As proposed by Spitzer–Fan, the refractive index is n = $\varepsilon^{\prime }$ and can be related to the density of states (ratio of free carrier to the effective mass, $N/m^{*}$) and the high frequency dielectric constant $\varepsilon_{\infty }$ [24,46,47],
(1)$$n^{2}=\varepsilon^{\prime }=\varepsilon_{\infty }-\frac{1}{4\pi^{2}\varepsilon_{0}}\left(\frac{e^{2}}{c^{2}}\right)\left(\frac{N_{c}}{m^{*}}\right)\lambda^{2}$$
where $m^{*}$ is the effective mass of each carrier. The $\varepsilon_{\infty }$ parameter can be determined from the extrapolation of the linear part to $\lambda^{2}=0$, as demonstrated by Figure 11. The estimated values of both $\varepsilon_{\infty }$ and $N_{c}/m^{*}$ are listed in Table 2. The value of $\varepsilon_{\infty }$ of PMMA–PVA films is found to be 2.623 and increases to 3.003 as the concentration of CuO NPs in PMMA–PVA/CuO nanocomposite is increased to 16%. The value of $\varepsilon_{\infty }$ is larger than n, confirming the existence of free charge carriers in PMMA–PVA/CuO nanocomposites [35,48].

Moreover, we investigate the relationship between (ε’’) and λ. The relaxation time (τ), optical mobility ($\mu_{opt})$, and optical resistivity ($\rho_{opt}$) can all be obtained from this relation as formulated by Drude free electron model [24],
(2)$$\varepsilon ’’=\frac{1}{4\pi^{3}\epsilon_{0}}\left(\frac{e^{2}}{c^{3}}\right)\left(\frac{N_{c}}{m^{*}}\right)\left(\frac{1}{\tau }\right)\lambda^{3}$$

Figure 12 shows the variation of ε’’ with $\lambda^{3}$ for the PMMA–PVA/CuO nanocomposites for various CuO NPs concentrations. τ can be evaluated from the slope of the plot of ε’’ versus $\lambda^{3}$ and from the value of $N_{c}/m^{*}$ and taking $m^{*}=0.44m_{e}$ [49]. The $\mu_{opt}$ and $\rho_{opt}$ of the films can be expressed as [24],
(3)$$\mu_{opt}=\frac{e\tau }{m^{*}}$$
(4)$$\rho_{opt}=\frac{1}{e\mu_{opt}N_{c}}$$

The calculated values of both parameters are presented in Table 2.

Table 3 presents a comparison between the values of key optical and optoelectronic parameters of PMMA–PVA/CuO NPs nanocomposites with the corresponding values of PMMA–PVA doped withTiO_2_ and SiO_2_ NPs reported in our previous works [17,19]. The most striking finding is simply evidenced by the ability to tune the values of the refractive index measured at 550 nm for optical device applications according to the type of NPs inserted homogenously into the polymeric matrix. Furthermore, careful inspection of the values of the extinction coefficient and the optical resistivity of PMMA–PVA/CuO NPs indicates that it absorbs light more efficiently than PMMA–PVA/TiO_2_ NPs and PMMA–PVA/SiO_2_ NPs nanocomposites. This has a direct impact on the fabrication of high-absorbing optoelectronic devices, such as organic solar cells and OLED devices. Therefore, our future work will be geared towards designing and fabricating multijunctional organic solar cells of projected high-conversion efficiency based on PMMA–PVA/CuO NPs nanocomposites carefully prepared under optimal synthesis conditions. Combining the results of this work and our previously reported findings would establish a good arena for interested research teams seeking optical and optoelectronic parameters accurately calculated.

### 3.3. Fourier Transform Infrared Spectroscopy (FTIR)

We employ the FTIR technique to elucidate the mechanism of bonding and to identify the vibrational bands of PMMA–PVA/CuO nanocomposites using Fourier transform infrared spectroscopy (FTIR) (Bruker Tensor 27 spectrometer, BRUKER CORPORATION, Billerica, Massachusetts, USA). The positions of the peaks of the FTIR patterns are found to shift due to the incorporation of CuO NPs into PMMA–PVA polymeric thin films. Figure 13 shows FTIR spectra of PMMA–PVA/CuO nanocomposite thin films with wt.% = 0%, 2%, 4%, 8%, and 16% of CuO NPs incorporated for a wavenumber range 500–4000 cm^–1^. Figure 14a shows FTIR spectra of PMMA–PVA/CuO nanocomposite thin films for a wavenumber range 500–1500 cm^−1^. Clearly, the FTIR spectra have three peaks located at 745, 1225, and 669 cm^−1^. The band at 669 cm^−1^ is assigned to the strong bending stretching vibration mode of (C–O) [27]. Additionally, the band at 745 cm^−1^ corresponds to the vibration mode of (C–H) [27]. Furthermore, the band at 1225 cm^−1^ is associated with the strong stretch bending vibration mode of the (C–O) bond [50]. The striking observation of FTIR spectra of the as-grown thin films is the broadening and shifting to a higher wavenumber as CuO NPs contents into PMMA–PVA polymeric thin films is increased. The new band of PMMA–PVA/CuO nanocomposite thin films is located at 1150 cm^−1^. The band at 1150 cm^−1^ is attributed to the stretching and symmetric bending vibration that may be assigned to the Cu–O bond [1]. Figure 14b shows FTIR spectra of PMMA–PVA/CuO nanocomposite thin films for a spectral range of 1500–2500 cm^−1^. Clearly, the FTIR spectra have one peak at 1727 cm^−1^. The broad band at 1727 cm^−1^ is attributed to the stretching modes of the carbonyl (C=O) group [1,50]. Figure 14c shows the FTIR spectra of PMMA–PVA/CuO nanocomposite thin films in the spectral range 2500–4000 cm^−1^. Clearly, the FTIR spectra have two peaks at 3020 and 2952 cm^−1^. The peak at 3020 cm^−1^ is associated with the presence of the free hydroxyl (OH) groups [1,45]. The peak at 2952 cm^−1^ is associated with the strong stretching mode of the (C–H) bond [1]. This can be interpreted in terms of the change in the strength of the bonds and the formation of new bonds that lead to the broadening and shifting of the peaks. The intensity of the peaks at 1727 and 2952 cm^−1^ are observed upon increasing CuO NPs doping levels. The reduction of the intensities of the peaks of the entire FTIR spectra is expected, and it could be attributed to the increase of the proportion of inorganic nanofiller into PMMA–PVA/CuO nanocomposite thin films.

Figure 15 shows FTIR spectra of PMMA–PVA polymeric thin films as a function of wavenumber in the range of 500–4000 cm^–1^. Figure 16 shows FTIR spectra of PMMA–PVA polymeric thin films in the ranges 500–1500 and 1500–4000 cm^−1^. Our results indicate that vibrational frequencies of PMMA–PVA polymeric thin films coincide with several vibrational frequencies observed for pure PVA polymer thin films. This observation is attributed to the fact that PVA thin film has dense molecular packing in the polymeric nanocomposite and stronger intermolecular hydrogen bonds in comparison with the PMMA component. The weak intermolecular hydrogen bonds of PMMA are responsible for the disappearance of functional groups of PMMA in the polymer blend [51]. Furthermore, the obtained FTIR spectrum demonstrates shifts of some band positions and changes in the intensities of some other bands compared with pure PMMA and PVA polymeric thin films. These shifts and the increases of intensities of some bands are due to the strong intramolecular interactions in PMMA–PVA blends. For PMMA–PVA polymer film, the vibrational band at 667 cm^–1^ could be ascribed to the (C–O) bending vibration. Whereas the vibrational band located at 743 cm^–1^ could be attributed to the bending vibration mode of the (C–H) bond. Additionally, the vibrational band positioned at 1213 cm^–1^ may be credited to the (C–O) bond stretching. Moreover, vibrational bands at 1713 cm^–1^ could be assigned to the stretching modes of carbonyl (C=O) groups [51]. Lastly, the broad band at 3020 cm^−1^ indicates the presence of the (C–H) stretching vibration. The increase and decrease of the peak intensities of the whole FTIR spectra of PMMA–PVA polymeric thin films are mainly due to the intermolecular bonding between the PMMA and PVA.

### 3.4. Thermogravimetric Analysis (TGA)

The PMMA–PVA/CuO nanocomposite thin films thermal stability is investigated by employing thermogravimetric (TGA) analysis for temperatures up to 400 °C using (TGA) (NETZSCH Premier Technologies, Exton, PA, USA) Figure 17 shows TGA thermograms of PMMA–PVA/CuO nanocomposite thin films containing different concentrations of CuO NPs. TGA thermograms of PMMA–PVA/CuO nanocomposite thin films show considerable weight loss (WL) steps at different temperatures. The TGA thermograms of PMMA–PVA/CuO nanocomposite thin films have two WL steps at 110 and 250 °C regardless of the degree of incorporation of CuO NPs. First and second WL are insignificantly shifted toward lower and higher temperatures indicating the influence of the change of intermolecular/intramolecular bonding. Clearly, the weight loss of pure PMMA–PVA polymeric thin films is observed to be about 92% at 400 °C. This value decreases as CuO NPs content in the polymeric matrix is increased. It attains a minimum value of 45% as the CuO NPs concentration in the polymeric film is increased to 16%. The WL of the nanocomposite is inversely proportional to their wt.% of CuO NPs content. This is a strong sign of the strengthening of physicochemical bonding density upon increasing the incorporation degree of CuO NPs in polymeric thin films. Conveniently, PMMA–PVA/CuO nanocomposite thin films are found to be thermally stable at temperatures below 110 °C. Most of the optical applications can be performed below this temperature in spite of the slight and negligible slope in the TGA thermograms below 110 °C, which is mostly due to water/solvent adsorption and can be easily tackled. The as-grown thin films have large photothermal conversion performance as employed in solar energy storage [27].

### 3.5. Scanning Electron Microscope (SEM)

The surface morphology of thin films is inspected using Scanning electron spectroscopy (SEM) (Hitachi High-Tech America Inc., Life Sciences and Nanotechnology - Dallas, TX, USA). The surface morphologies of PMMA–PVA/CuO nanocomposite thin films at different concentrations of CuO NPs at 5 μm magnification are shown in Figure 18. Figure 18a shows that the nanocomposite thin films of PMMA–PVA exhibit an amorphous nature with a smooth surface. Figure 18b,c show the morphology of the polymer nanocomposites containing 2% and 4% of CuO NPs. The CuO NPs have a spherical shape, and nanoparticles sizes range from 50 to 100 nm. As the concentration of CuO NPs increases to 8% and 16%, smooth ganglia-like hills with some wrinkles (spherical longitudinal shapes) are observed, as demonstrated by Figure 18d,e. This could be attributed to a substantial increase in the lamellar twisting period and a decreased radial growth rate in amorphous regions within the polymeric blend. It indicates some crystalline domains with coarse spherulitic structures. This is due to CuO NPs segregated into interlamellar or intercrystallite regions of the blend. For PMMA–PVA/CuO 8%, CuO NPs are weakly linked within the polymeric matrix. As the concentration of CuO NPs is doubled, NPs exhibit rod-like linkage to the constituents of the polymeric matrix. Furthermore, SEM was used to examine the morphology and dispersion of CuO NPs on the surface of PMMA–PVA films. The SEM images show a good dispersion of CuO NPs on the surface of the PMMA–PVA films. This provides substantial evidence of the validity of our synthesis process of obtaining CuO NPs.

## 4. Conclusions

In summary, PMMA–PVA/CuO nanocomposite thin films with different CuO NPs content in the range of 0% to 16% are synthesized on glass substrates using the dip-coating technique. Optical properties of the synthesized nanocomposite thin films, including transmittance $\left(T\%\right)$, reflectance $\left(R\%\right)$, absorption coefficient $\left(\alpha \right)$, optical constants (n and k), and optical dielectric functions ($\varepsilon_{1}$ and $\varepsilon_{2}$) are calculated and interpreted using experimental transmittance and reflectance spectra. Moreover, a combination of classical models, such as Tauc, Urbach, Spitzer–Fan, and Drude models, are applied to deduce the optical, optoelectronic parameters, and the energy gaps of as-prepared nanocomposite thin films. Calculated refractive indices (*n*) of undoped PMMA–PVA film are found to lie in the range 1.5–1.85. The optical band gap of PMMA–PVA polymeric thin film is found to be 4.101 eV. This value decreases as CuO NPs are introduced into polymer thin films. The obtained high refractive index values of nanocomposite thin films indicate their potential for strong optical confinement applications and enhance the optical intensities of nonlinear interactions. The high transmittance, wide band gap energy, and high refractive index of the nanocomposite films indicate that doped polymerized thin films could be potential candidates for optoelectronic devices. To elucidate and identify different vibrational bands associated with major bonds of the nanocomposite thin films, FTIR transmittance spectra of as-grown thin films are studied and interpreted in the spectral range 500–4000 cm^−1^. Investigating thermal stability is very important. The TGA thermograms of as-prepared doped polymerized thin films show that the weight loss of the nanocomposite is inversely proportional to the wt.% of CuO NPs. This confirms that physicochemical bonding density is strengthened by incorporating a high concentration of CuO NPs. Expediently, PMMA–PVA/CuO thin films are thermally stable at temperatures below 110 °C. Interestingly, the vast majority of optical applications operate below 110 °C. A careful comparison of the values of optical and optoelectronic parameters of PMMA–PVA polymeric thin films doped with TiO_2_ and SiO_2_ with the corresponding values of PMMA–PVA/CuO NPs nanocomposite reveals that it is easy to tune the values of the refractive index measured at 550 nm and other key optical parameters for a diversity of optoelectronic applications by appropriately selecting the type of inorganic NPs. Moreover, the values of the extinction coefficient and the extremely large value of the optical mobility of (PMMA–PVA)/CuO NPs show that it exhibits outstanding absorbance that subsidizes the low absorbance of PMMA–PVA/TiO_2_ NPs and PMMA–PVA/SiO_2_ NPs nanocomposites. This allows the manufacturing of high-absorbing optoelectronic devices, such as organic solar cells and OLED devices. Having synthesized organic–inorganic blends of a wide range of optoelectronic properties, we can think of designing and fabricating high-efficiency multijunctional organic solar cells. Combining the results of this work and optical parameters of PMMA–PVA/TiO_2_ NPs and PMMA–PVA/SiO_2_ NPs nanocomposites would establish a solid for future research on organic–inorganic nanocomposites.

Such detailed investigation of organic–inorganic hybrid polymerized thin films would lead to the fabrication of optimized functionality.

## Figures and Tables

**Figure 1 polymers-13-01715-f001:**
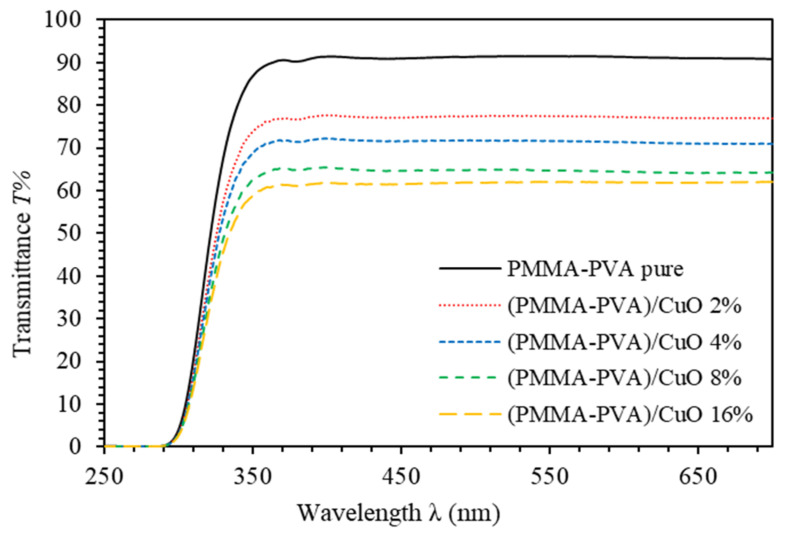
Transmittance spectra of PMMA–PVA/CuO nanocomposite thin films as a function of wavelength containing different CuO NPs concentrations.

**Figure 2 polymers-13-01715-f002:**
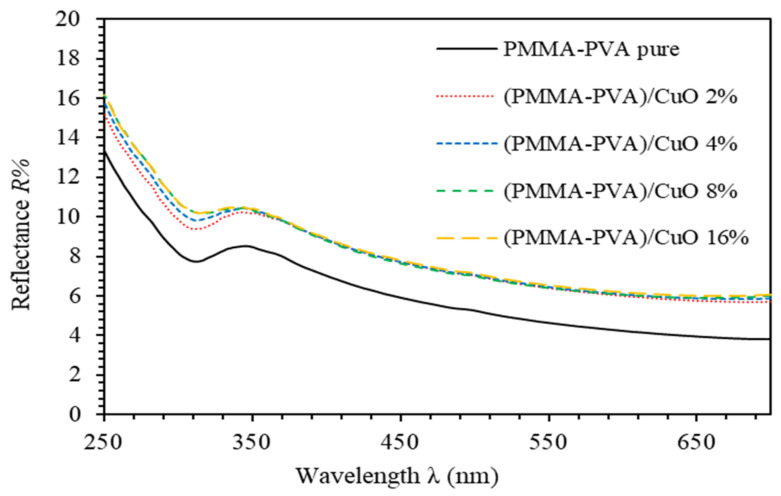
Reflectance spectra of PMMA–PVA/CuO nanocomposite thin films as a function of wavelength containing various CuO NPs concentrations.

**Figure 3 polymers-13-01715-f003:**
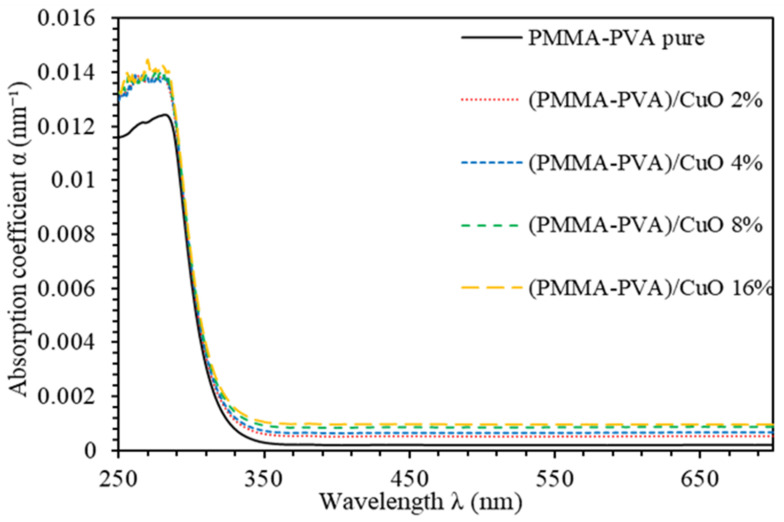
Absorption coefficient spectra (α) of PMMA–PVA/CuO nanocomposite thin films as a function of wavelength for different CuO NPs concentrations.

**Figure 4 polymers-13-01715-f004:**
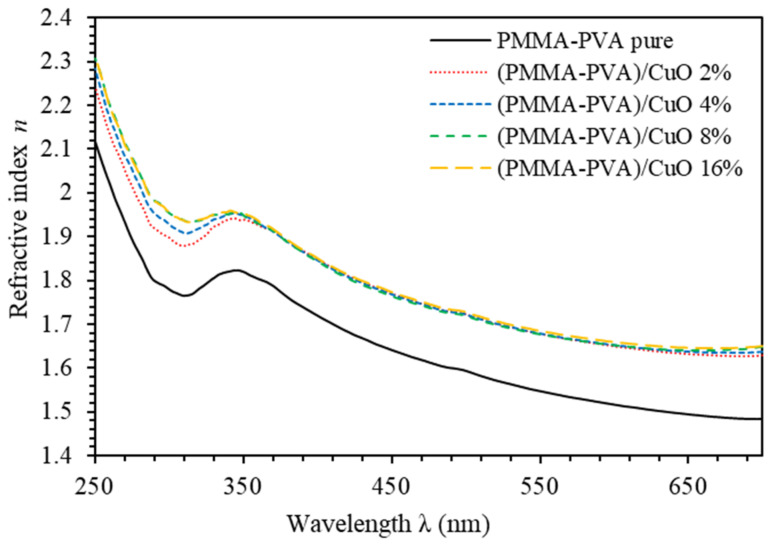
Refractive index spectra of PMMA–PVA/CuO nanocomposite thin films as a function of wavelength for various CuO NPs concentrations.

**Figure 5 polymers-13-01715-f005:**
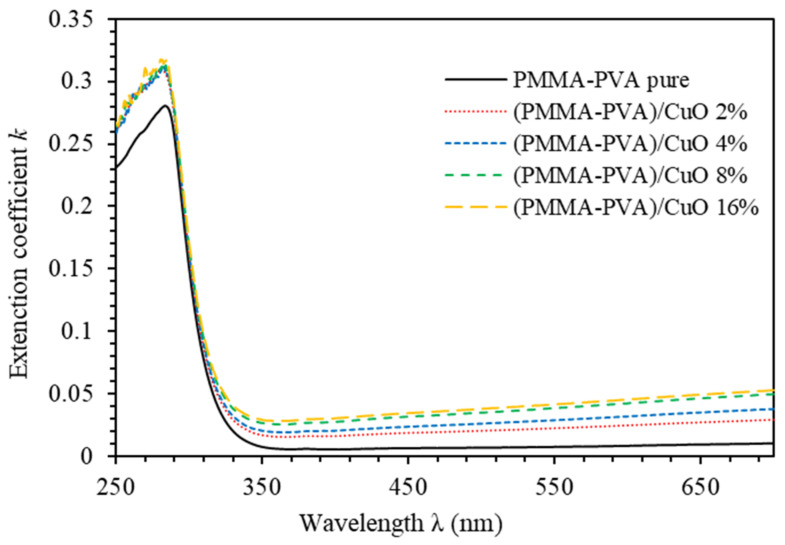
Extinction coefficient spectra of PMMA–PVA/CuO nanocomposite thin films as a function of wavelength for various CuO NPs concentrations.

**Figure 6 polymers-13-01715-f006:**
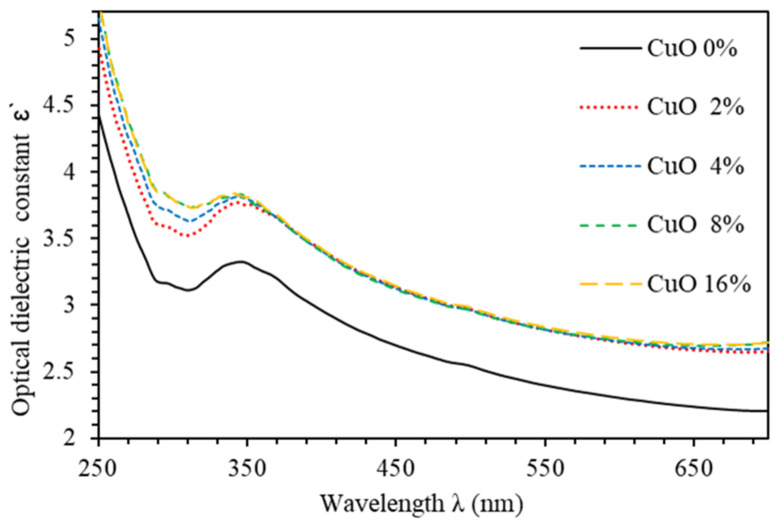
Optical dielectric constant (ɛ′) spectra of PMMA–PVA/CuO nanocomposite thin films as a function of wavelength for various CuO NPs concentrations.

**Figure 7 polymers-13-01715-f007:**
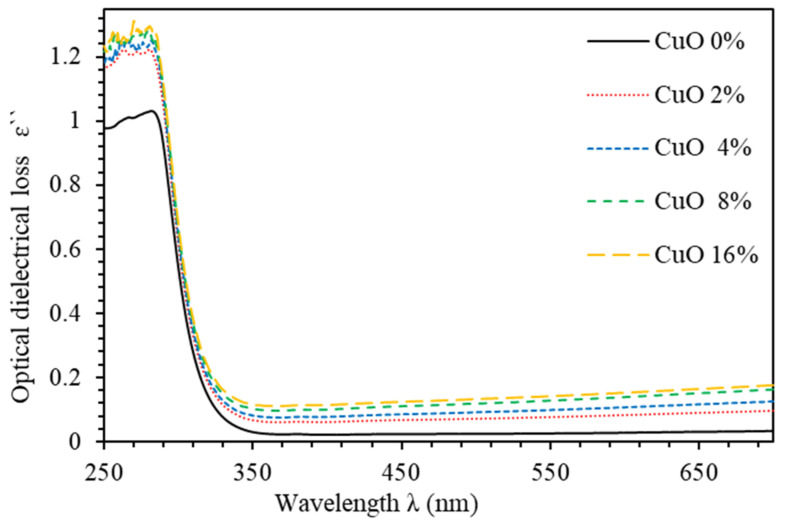
Optical dielectric loss (ɛ′’) spectra of PMMA–PVA/CuO nanocomposite thin films as function of wavelength for various CuO NPs concentrations.

**Figure 8 polymers-13-01715-f008:**
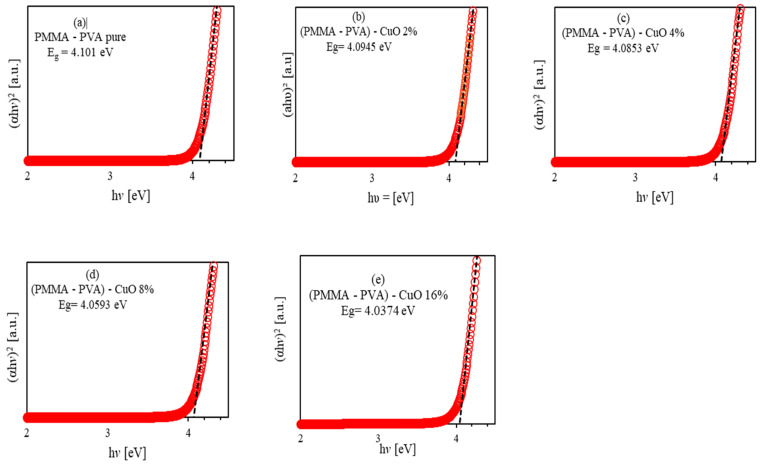
Tauc optical bandgap energy of PMMA–PVA/CuO nanocomposite thin films as a function of wavelength for various CuO NPs concentrations deposited by the dip-coating technique, (**a**) band gap of undoped PMMA–PVA polymer, (**b**) band gap of PMMA–PVA/CuO 2%, (**c**) band gap of PMMA–PVA/CuO 4%, (**d**) band gap of PMMA–PVA/CuO 8%, and (**e**) band gap of PMMA–PVA/CuO 16%.

**Figure 9 polymers-13-01715-f009:**
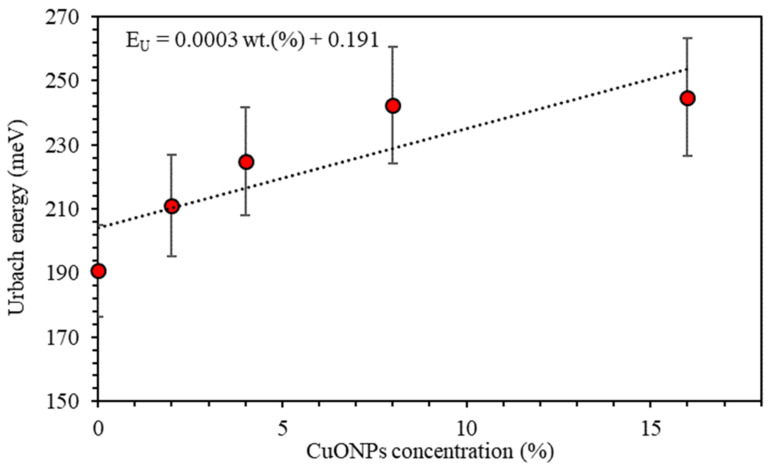
Urbach energy versus the concentration of CuO NPs in the PMMA–PVA/CuO nanocomposite thin films.

**Figure 10 polymers-13-01715-f010:**
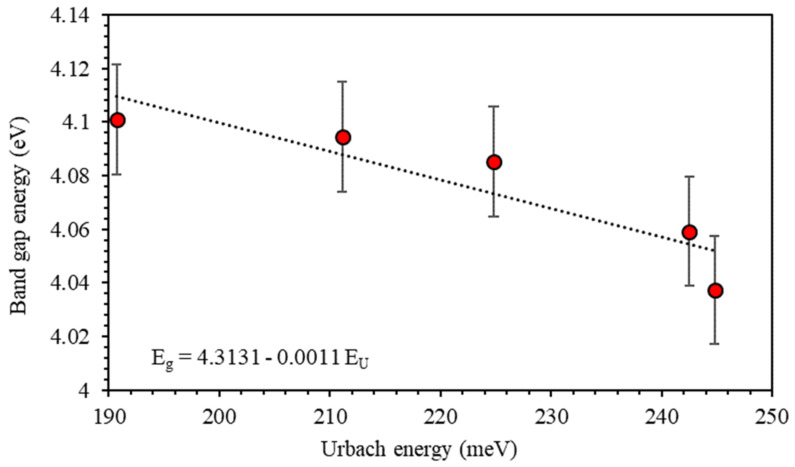
The relationship between the optical energy gap and the Urbach energy of PMMA–PVA/CuO nanocomposite thin films.

**Figure 11 polymers-13-01715-f011:**
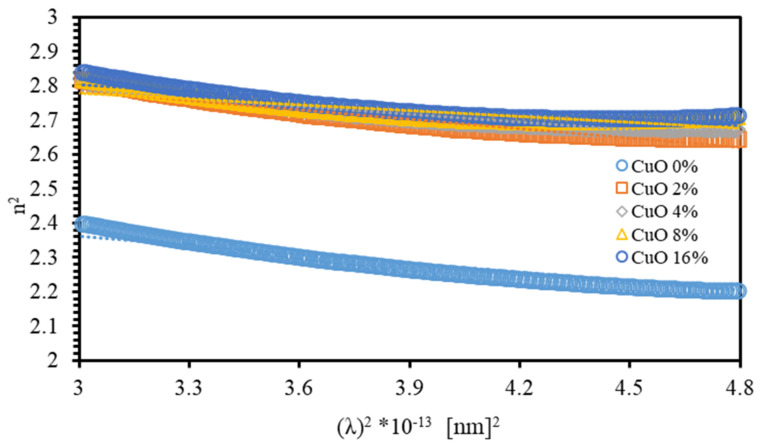
The variation of the real part of the dielectric constant ($n^{2}=\varepsilon^{\prime }$) with the square of the photon wavelength ($\lambda^{2}$ ) for PMMA–PVA/CuO nanocomposites.

**Figure 12 polymers-13-01715-f012:**
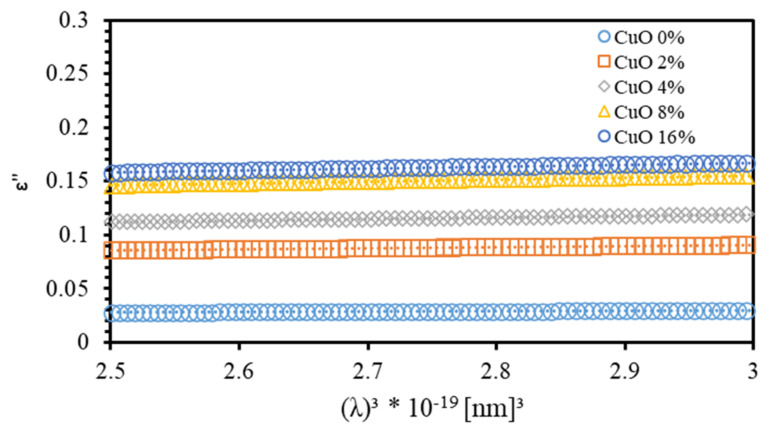
The variation of the spectra of the imaginary part (ε’’) of PMMA–PVA/CuO nanocomposite thin films as a function of (λ^3^) for various CuO NPs concentrations.

**Figure 13 polymers-13-01715-f013:**
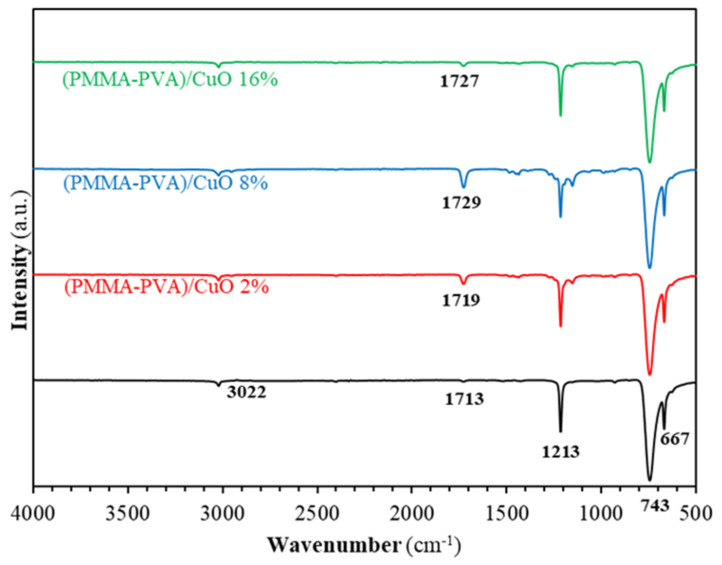
FTIR spectra of doped and undoped for PMMA–PVA/CuO nanocomposite thin films.

**Figure 14 polymers-13-01715-f014:**
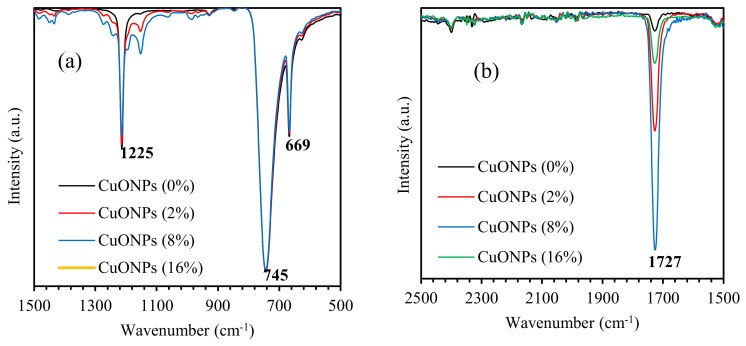
FTIR spectra of PMMA–PVA/CuO nanocomposite thin films in the spectral ranges: (**a**) 500–1500 cm^−1^, (**b**) 1500–2500 cm^−1^, and (**c**) 2500–4000 cm^−1^.

**Figure 15 polymers-13-01715-f015:**
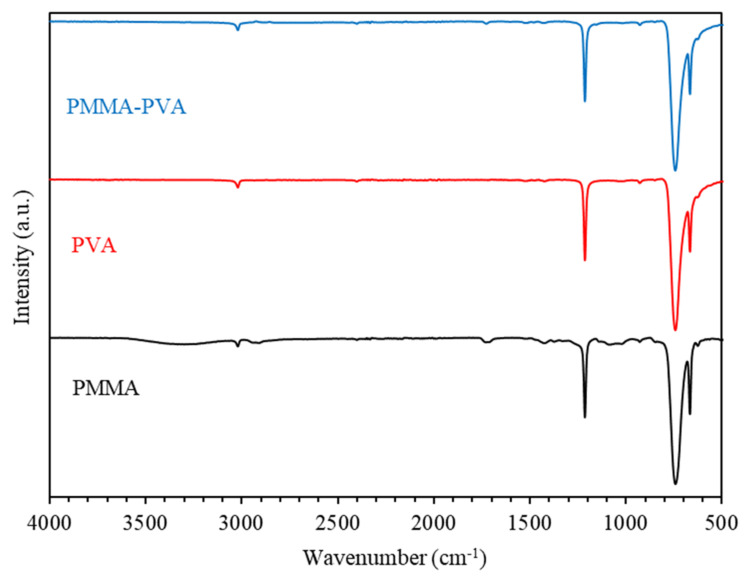
FTIR spectra of PMMA–PVA polymeric thin films as a function of wavenumber.

**Figure 16 polymers-13-01715-f016:**
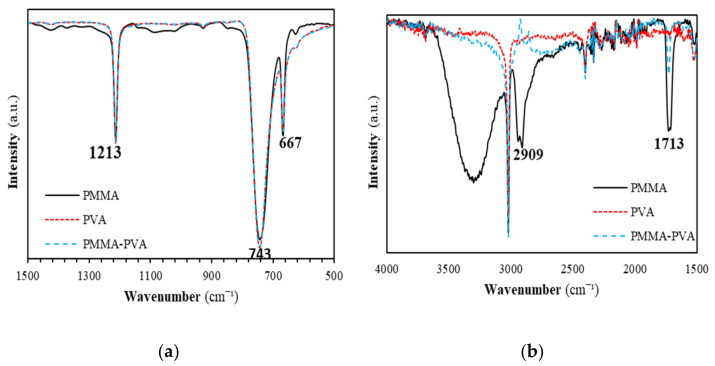
FTIR spectra of PMMA–PVA polymeric thin films in the spectral range (**a**) 500–1500 cm^−1^ and (**b**) 1500–4000 cm^−1^.

**Figure 17 polymers-13-01715-f017:**
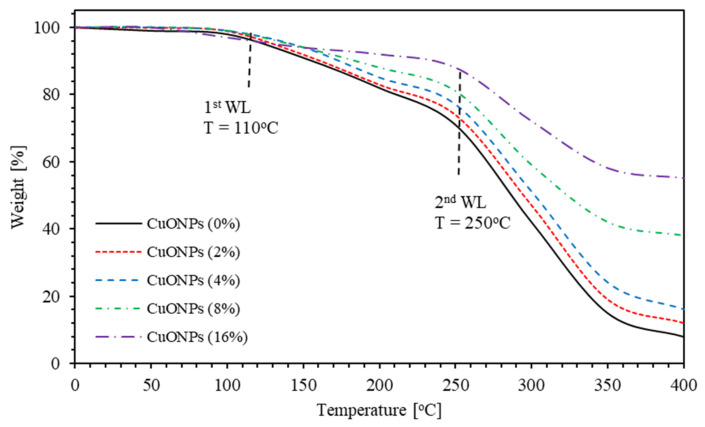
TGA thermograms of PMMA–PVA/CuO nanocomposite thin films for various CuO NPs concentrations.

**Figure 18 polymers-13-01715-f018:**
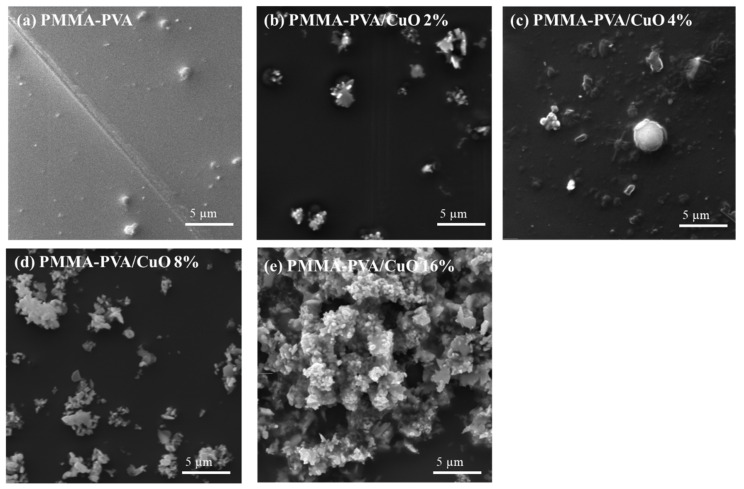
The SEM micrographs of PMMA–PVA/CuO NPs nanocomposite thin films at different concentrations of CuO NPs, (**a**) undoped PMMA–PVA polymer, (**b**) PMMA–PVA/CuO 2% NPs nanocomposite, (**c**) PMMA–PVA/CuO 4% NPs nanocomposite, (**d**) PMMA–PVA/CuO 8% NPs nanocomposite, and (**e**) PMMA–PVA/CuO 16% NPs nanocomposite.

**Table 1 polymers-13-01715-t001:** The refractive index (*n*), extinction coefficient (*k*), dielectric permittivity (*ε′*), and dielectric loss (*ε″*) of PMMA–PVA/CuO NPs thin films for different concentrations of CuO NPs at λ = 550 nm.

CuO NPs (wt.%)	Refractive Index *n*	Extinction Constant *k*	Dielectric Permittivity $$\varepsilon^{\prime }$$	Dielectric Loss $$\varepsilon^{\prime \prime }$$
0	1.548	0.0077	2.396	0.0238
2	1.678	0.0224	2.814	0.0752
4	1.679	0.0291	2.816	0.0977
8	1.678	0.0381	2.814	0.1278
16	1.685	0.0417	2.836	0.1404

**Table 2 polymers-13-01715-t002:** The estimation of key essential optoelectronic parameters of PMMA–PVA/CuO nanocomposite thin films for various CuO NPs concentrations.

Parameter	CuO 0%	CuO 2%	CuO 4%	CuO 8%	CuO 16%
Density of states, $N_{c}/m^{*}*\;10^{+57}$ ($m^{-3}.Kg^{-1}$)	1.066	1.116	0.944	0.691	0.812
Charge carrier density, $N_{c}*\;10^{+27}$ ($m^{-3}$)	4.273	4.472	3.782	2.770	3.255
High frequency dielectric constant, $\varepsilon_{\infty }$	2.623	3.058	3.017	2.948	3.003
Relaxation time, $\tau *10^{-14}$ (s)	2.493	0.987	0.599	0.342	0.402
Optical mobility, $\mu_{opt}*10^{-3}$	9.951	3.938	2.391	1.364	1.603
Optical resistivity, $\rho_{opt}\;*10^{-6}$	1.470	3.548	6.910	16.548	11.979

**Table 3 polymers-13-01715-t003:** A comparison of key essential optical and optoelectronic parameters of PMMA–PVA doped with 16% wt. of TiO_2_, SiO_2_, and CuO NPs.

Parameter	PMMA–PVA	PMMA–PVA/TiO_2_	PMMA–PVA/SiO_2_	PMMA–PVA/CuO
Refractive index (at 550 nm)	1.548	2.137	1.898	1.685
Extenction coefficient (at 550 nm)	0.0077	0.0181	0.0165	0.0417
Optical Band gap (eV)	4.101	4.050	4.047	4.037
$\mathrm{Density}\;\mathrm{of}\;\mathrm{states},\;N_{c}/m^{*}*\;10^{+57}$ ($m^{-3}.Kg^{-1}$)	1.066	3.125	1.415	0.812
High frequency dielectric constant, $\varepsilon_{\infty }$	2.623	5.268	3.917	3.003
$\mathrm{Optical}\;\mathrm{mobility},\;\mu_{opt}*10^{-3}$	9.951	12.675	5.707	1.603
$\mathrm{Optical}\;\mathrm{resistivity},\;\rho_{opt}\;*10^{-6}$	1.470	0.394	1.932	11.979

## Data Availability

Not applicable.

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
