# Peer review of "Synthesis of Optically Tunable and Thermally Stable PMMA–PVA/CuO NPs Hybrid Nanocomposite Thin Films"

_polymers, 2021, doi:10.3390/polym13111715_

Round 1

Reviewer 1 Report

  1. The abstract is too long and poorly organized.
  2. Lack of novelty.
  3. Similar works relevant to this work should be presented and discussed, and the advancement of this work was not clearly demonstrated in Introduction.
  4. The detailed information of the materials used in this work is not presented.
  5. The figure format need to be improved.
  6. What did the authors want to imply for presenting the FTIR results?

Author Response

Dear reviewer,

We have addressed all the comments, questions, and remarks raised by you. Please refer to the attachment replies.

Please accept our kind regards

Reviewer 2 Report

The authors present an extensive study about the synthesis and characterization of PMMA-PVA/CuO NPs, giving explanations accesible to a broader audience. However, there are some issues that must be addressed:

  1. The originality of the study is not well motivated, in comparison, firstly, with their other published articles and secondly, with the work of other researchers. The authors have similar studies in which PMMA-PVA/TiO2 NPs or PMMA-PVA/SiO2 NPs were investigated.
  2. It must be specified the devices used for the characterization of samples.
  3. The SEM images are at a low resolution and must be changed with images of a higher quality.
  4. The manuscript has to be put in the journal's format.

Author Response

Dear reviewer,

We have addressed all questions, remarks, and comments raised by you. Please find attached our replies.

Please accept our regards,

Round 2

Reviewer 1 Report

I still feel this article is not innovative enough to be published in the Journal of Polymers.

Author Response

Dear Reviewer,

We have addressed all your concerns about the novelty of the work. Please see attached our replies.

Please accept our regards,

Prof. Dr. Ahmad ALsaad
